# DIA-Based Quantitative Proteomics Reveals the Protein Regulatory Networks of Floral Thermogenesis in *Nelumbo nucifera*

**DOI:** 10.3390/ijms22158251

**Published:** 2021-07-31

**Authors:** Yueyang Sun, Yu Zou, Jing Jin, Hao Chen, Zhiying Liu, Qinru Zi, Zeyang Xiong, Ying Wang, Qian Li, Jing Peng, Yi Ding

**Affiliations:** 1State Key Laboratory of Hybrid Rice, Department of Genetics, College of Life Sciences, Wuhan University, Wuhan 430072, China; yueyangsun@whu.edu.cn (Y.S.); zouxiaoyu@whu.edu.cn (Y.Z.); zhiying-liu@whu.edu.cn (Z.L.); qinruzi@whu.edu.cn (Q.Z.); zeyangxiong@whu.edu.cn (Z.X.); yingwang@whu.edu.cn (Y.W.); liqian0074@whu.edu.cn (Q.L.); 2Department of Biotechnology, College of Life Sciences, Guizhou University, Guiyang 550025, China; jinjing1130@whu.edu.cn; 3Institute of Vegetables, Wuhan Academy of Agricultural Sciences, Wuhan 430065, China; haochen2013@whu.edu.cn (H.C.); pengjing67@163.com (J.P.)

**Keywords:** floral thermogenesis, *Nelumbo nucifera*, DIA-based quantitative proteomics, time series analysis, protein-protein interaction network, WGCNA

## Abstract

The sacred lotus (*Nelumbo nucifera*) can maintain a stable floral chamber temperature between 30 and 35 °C when blooming despite fluctuations in ambient temperatures between about 8 and 45 °C, but the regulatory mechanism of floral thermogenesis remains unclear. Here, we obtained comprehensive protein profiles from receptacle tissue at five developmental stages using data-independent acquisition (DIA)-based quantitative proteomics technology to reveal the molecular basis of floral thermogenesis of *N. nucifera*. A total of 6913 proteins were identified and quantified, of which 3513 differentially abundant proteins (DAPs) were screened. Among them, 640 highly abundant proteins during the thermogenic stages were mainly involved in carbon metabolism processes such as the tricarboxylic acid (TCA) cycle. Citrate synthase was identified as the most connected protein in the protein-protein interaction (PPI) network. Next, the content of alternative oxidase (AOX) and plant uncoupling protein (pUCP) in different tissues indicated that AOX was specifically abundant in the receptacles. Subsequently, a protein module highly related to the thermogenic phenotype was identified by the weighted gene co-expression network analysis (WGCNA). In summary, the regulation mechanism of floral thermogenesis in *N. nucifera* involves complex regulatory networks, including TCA cycle metabolism, starch and sucrose metabolism, fatty acid degradation, and ubiquinone synthesis, etc.

## 1. Introduction

Body heat production is generally considered to occur in homeothermic animals, which can adapt to changes in environmental temperature through thermoregulation [1]. However, similar thermogenesis has also been found and described in the floral organs of some flowering plants, including Araceae [2,3,4,5], Nelumbonaceae [6], Nymphaeaceae [7], Magnoliaceae [8], Cycadaceae [9], and so on. The appropriate temperature generated by thermogenesis contributes to the prevention of the flower organs from suffering low temperatures [3], promoting the success of double fertilization [10], facilitating pollen germination and pollen tube growth [11], and attracting insect pollinators by emitting odors and providing heat rewards [12,13].

With the deepening of research on thermogenesis in plants, the molecular mechanism of floral thermogenesis has been gradually explored. There are two energy-dissipating systems thought to be related to the thermogenesis of plants [14,15]. One is the alternative pathway of plant mitochondria mediated by alternative oxidases (AOXs) and the other involves plant uncoupling proteins (pUCPs). In the alternative pathway, electrons that bypass cytochrome c reductase (complex III) and cytochrome c oxidase (COX, complex IV) are directly transferred from ubiquinone to AOX and then transferred to oxygen through AOX. Therefore, most of the energy is released as heat with no energy conservation for the electron flux through the alternative pathway [16]. AOX protein is a nuclear-encoded mitochondrial inner membrane protein that functions as the terminal oxidase of the alternative pathway [17]. In thermogenic plants, the AOX activity is related to thermogenesis, which can characterize the capacity of the alternative pathway and increases in thermogenic tissue when thermogenesis occurs [18,19]. For example, the increased AOX activity was detected during the thermogenic stages in *Nelumbo nucifera*, while it returned to the initial level when thermogenesis ended [20].

Uncoupling proteins (UCPs) form a subfamily of mitochondrial carrier protein, which are located on the inner membrane of mitochondria, and can dissipate the ΔμH^+^ generated by the respiratory chain so that protons can bypass the ATP synthase and be transported from the membrane space to the mitochondrial matrix [21,22,23]. This process leads to the uncoupling of respiration and phosphorylation, and electrochemistry energy is released as heat [23]. It was found that pUCPs might participate in thermogenesis in the thermogenic plant, such as skunk cabbage (*Symplocarpus renifolius*). SrUCPA, the main pUCPs of *S. renifolius*, had a high abundance in spadix mitochondria, meanwhile SrUCPA and SrAOX were specifically co-expressed in the thermogenic tissue or stage, which indicated that they might play a role in the thermogenesis of skunk cabbage [24]. 

*N. nucifera*, an important aquatic vegetable crop that blooms in early summer, is also a thermogenic plant that has been described in some detail [25]. The relationship between growth development and the thermogenic period of *N. nucifera* flowers can be divided into three distinct physiological phases: Pre-thermogenic, thermogenic, and post-thermogenic phases. Thermogenesis mainly occurs in the receptacle tissue with increased activity at the thermogenic phase and cessation at the post-thermogenic phase [20]. *N. nucifera* is the only thermogenic plant to date that the electron flux flowing to AOX (AOX flux) and COX (COX flux) has been quantified *in vivo* using the stable oxygen isotope technique [20,26]. Measurement of the electron flux through the AOX pathway can accurately determine AOX activity [27]. The AOX flux increased significantly during the thermogenic stages and could even account for 93% of the total respiratory flux in the hottest receptacles, while the COX flux did not change significantly [20]. There was a significantly positive correlation between the thermogenesis levels in thermogenic receptacles and both total respiratory flux and AOX flux, but there was almost no correlation with COX flux [20]. Therefore, AOX was considered to play a role in the regulation of the floral thermogenesis of *N. nucifera*. Unlike *Sauromatum guttatum* [28] and *Arum italicum* [19], the increase of AOX protein in *N. nucifera* receptacles was synchronized with the increase in thermogenic activity [20]. Amino acid multiple sequence alignment and phylogenetic analysis showed that no specific sequence of AOX protein related to thermogenic activity was found in *N. nucifera* [27]. Similarly, no specific primary structures of AOXs and pUCPs related to thermogenesis were found in *S. renifolius* [29]. Hence, there are probably additional factors involved in the regulation of floral thermogenesis. Recently, with the development of high-throughput sequencing technology, several transcriptome-level studies have been performed to reveal the molecular basis of floral thermogenesis. In *S. renifolius*, the utilization of SuperSAGE sequencing revealed gene expression profiles related to the maintenance and termination of thermogenesis [30]. In *Arum concinnatum*, 1266 genes highly correlated with the temperature trend of the flower organs were identified by transcriptome analysis [31]. Then, integrated analysis of small RNA and transcriptome sequencing also revealed the role of miRNAs (microRNAs) in the regulation of the floral thermogenesis of *N. nucifera* [32]. However, so far, there have been no proteomic studies on plant thermogenesis that have been reported.

Data-independent acquisition (DIA)-based quantitative proteomics technology is an emerging technology for large-scale protein identification and quantification in recent years. It has the advantages of deep proteome coverage, high quantitative reproducibility, and accuracy, and has been applied in biomarker research, clinical research, basic research, and other fields [33,34,35]. In the present study, we provided comprehensive protein profiles from receptacle tissue using DIA-based quantitative proteomics technology to further reveal the molecular basis of floral thermogenesis in *N. nucifera*. The protein regulatory networks related to thermogenesis were identified based on the correlation between protein abundance and the thermogenesis pattern (Appendix A). These global protein profiles during floral thermogenesis in *N. nucifera* have provided new insights into plant thermogenesis and lays a solid foundation for in-depth study on the regulatory mechanisms of floral thermogenesis.

## 2. Results

### 2.1. An Overview of the Quantitative Proteomics Analysis

The experimental strategy of data-independent acquisition mass spectrometry (DIA-MS) identification technology in the present study is shown in Appendix A. DIA-MS analysis requires the construction of a data-dependent acquisition (DDA)-based spectral library before protein quantification. As a result, 44976 peptides corresponding to 9461 proteins were identified in the spectral library (Appendix A). The DIA data with the spectra library were analyzed by Spectronaut Pulsar 11.0 (Biognosys AG, Switzerland), and the results showed that 6913 proteins were further quantified (Appendix A). After being mapped to multiple public databases, the 6913 identified proteins were annotated (Appendix A). Statistics of the proteins and peptides identified for each sample by DIA-MS are shown in Figure 1A. 

To assess the quality of the proteomic data, quality control (QC) analysis was performed. The median of intra-group coefficient of variation (CV) distribution for each sample was less than 10% (Figure 1B). Principle component analysis (PCA) showed that the samples from five different stages could be well separated, and samples from the same stage could be clustered well (Figure 1C). Pearson correlation coefficient of the protein levels between every two samples showed that three replicates in the same stages had a high correlation (Appendix A). All of these analyses indicated the good repeatability and reliable quality of the proteomic data in this study.

Proteins with a fold change (FC) ≥ 2 and *q*-value < 0.05 were defined as significantly differentially abundant proteins (DAPs). In total, 3513 DAPs of ten comparison groups were identified in the present study (Figure 1D and Appendix A). The DAPs increased with the development of flowers when using stage 1 (S1) as a control. Moreover, the DAPs between adjacent stages (stage 2 (S2) vs. S1, stage 3 (S3) vs. S2, stage 4 (S4) vs. S3, stage 5 (S5) vs. S4) showed that S3 vs. S2 and S5 vs. S4 had more DAPs, which might be related to the larger developmental changes of *N. nucifera* flowers in these two comparative stages (from S2 to S3, the flowers went from buds to blooms, and from S4 to S5, the petals withered gradually). When using stage 3 as a control, the number of DAPs of the comparison groups between different thermogenic stages (S3 vs. S2 and S3 vs. S4) was less than the comparison groups between thermogenic and non-thermogenic stages (S3 vs. S1 and S3 vs. S5).

### 2.2. Time-Series Analysis of DAPs

To reveal the time dynamics of the abundance of all DAPs, we performed the time-series cluster analysis based on five continuous developmental stages. As a result, all DAPs were classified into six clusters with different expression patterns (Figure 2 and Appendix A). Among the six clusters, the proteins in cluster 3 (C3) which had a high abundance during thermogenesis increased at stage 2, reached a peak at stage 3 or stage 4, and decreased gradually at stage 5, while the proteins in cluster 5 (C5) had a low abundance in the thermogenic stages (Appendix A). The expression patterns of the proteins in C3 were basically consistent with the thermogenesis pattern while that of C5 were opposite. A total of 640 and 429 proteins were respectively classified into C3 and C5 (Appendix A). Considering that the expression patterns of the proteins in C3 and C5 were related to the thermogenesis pattern in the thermogenic receptacles during the five stages, we speculated that these two protein clusters were related to the thermogenic process, which occurred significantly from stage 2 to stage 4 (Appendix A).

### 2.3. Functional Annotation Analysis of the Protein Clusters Related to the Thermogenic Process

To further understand the function of the proteins related to thermogenesis in C3 and C5, the gene ontology (GO), Kyoto encyclopedia of genes and genomes (KEGG) pathway enrichment analysis, and Eukaryotic orthologous groups (KOG) annotation were performed (Appendix A). For the annotation of C3, the main significantly enriched GO terms of the biological process (BP) included tricarboxylic acid cycle (TCA cycle), aerobic respiration, and oxidation-reduction process (Appendix A). Oxidoreductase activity, catalytic activity, and pyruvate dehydrogenase activity were the main enriched GO terms of molecular function (MF). Moreover, the main representative GO terms of the cellular component (CC) were the TCA cycle enzyme complex, mitochondrion, and the dihydrolipoyl dehydrogenase complex. KEGG pathway enrichment results showed that citrate cycle, biosynthesis of secondary metabolites, carbon metabolism, and so on, were the most significantly enriched (Figure 3A, Appendix A). KOG annotation showed that many proteins were concentrated in energy production and conversion, carbohydrate transport and metabolism, and post-translational modification (Appendix A). For the annotation of C5, photosynthesis, signal transducer activity, and photosystem were the most enriched GO terms (Appendix A). Photosynthesis, plant hormone signal transduction, and MAPK signaling pathway were the top-three enriched KEGG pathways (Figure 3B, Appendix A). KOG annotation showed that signal transduction mechanisms covered the most proteins except for only the general function prediction (Appendix A).

AOX proteins that had been proposed to play an important role in the thermogenic process of *N. nucifera* [20,27] were identified in cluster 3, including NNU_06050-RA, NNU_06051-RA, and NNU_10976-RA (in the present study, NNU_06050-RA represents the corresponding protein ID of gene *NNU_06050*, as are other protein IDs). Photosynthesis, the most enriched pathway of cluster 5, was reported to be related to the function of the receptacles after the end of thermogenesis [36]. This indicated that the DAPs in cluster 3 were highly abundant during the thermogenic stages and might be closely related to floral thermogenesis in *N. nucifera*.

### 2.4. Validation of the Proteomic Data by qRT-PCR and Western Blot

To verify the expression patterns of the proteins identified by DIA-MS technology, nine randomly selected DAPs and three interested proteins were verified by qRT-PCR, which reflected their expression at the transcriptional level. It has been proposed that the protein AOX or pUCP participates in the thermogenesis of some thermogenic plants [20,24]. Our results indicated that AOX1a (NNU_06050-RA) and pUCPs (NNU_03736-RA, NNU_13276-RA) showed the same expression pattern between the relative expression level in qRT--PCR, fold changes in RNA--seq data (accessed from Zou et al. [32]), and the proteomic data (Figure 4A–D). Similarly, the expression patterns of the transcripts of seven DAPs related to the TCA cycle, carbon metabolism, and oxidative phosphorylation in cluster 3 were basically consistent with that of the RNA--seq data and proteomic data (Figure 4E–M). Although they all had almost the same expression pattern between the proteomic level, RNA-seq level, and qRT-PCR level, the proteomic abundance peaks of the above ten proteins basically lagged behind the corresponding transcripts expression peaks. However, the transcript of a protein (probably inactive leucine-rich repeat receptor-like protein kinase IMK2, NNU_06853-RA) in cluster 5 had a consistent expression change with RNA-seq data but not with proteomic data. Our qRT-PCR and RNA-seq results both showed that *NNU_06853* had the lowest expression at stage 5 while the proteomic data showed that NNU_06853-RA had the lowest abundance at stage 4, and its abundance went up in stage 5 (Figure 4L,N). This different expression pattern between protein level and transcript level might be caused by the post-transcriptional regulation of the transcripts. ATP synthase subunit delta, chloroplastic, NNU_09824-RA (ATPD), a protein in cluster 1 related to photosynthesis, had a high abundance at stage 5, and its corresponding transcript also had the same expression pattern at the qRT-PCR and RNA-seq levels (Figure 4L,O).

The expression patterns of three proteins, including AOX1a, pUCP1 (NNU_03736-RA), and catalase (CAT, NNU_23981-RA), were further validated by our Western blot analysis (Figure 5A–C). The expression patterns of the three proteins in the Western blot experiments (Figure 5A) were highly consistent with the proteomic data (Figure 5B). When using stage 1 as a control, the abundance of AOX increased significantly in the other stages, while only pUCP had a significant increase in stage 4, and there was no significant difference in the abundance of CAT (Figure 5C). Coincidentally, the significant analysis of the Western blot results was consistent with the DAPs analysis of the proteomic data. Overall, all the above results showed that our proteomic data were reliable.

### 2.5. Abundance of the Proteins AOX and pUCP between Thermogenic and non-Thermogenic Tissues

To further understand the abundance of AOX and pUCP in different tissues, we detected the content of these two proteins between thermogenic receptacles and non-thermogenic leaves of *N. nucifera*. Total protein was extracted and analyzed by Western blot using anti-AOX and anti-UCP antibodies (Figure 5D). The expression patterns of AOX and pUCP were opposite when compared between thermogenic receptacles and non-thermogenic leaves. AOX was extremely abundant in the thermogenic receptacles during thermogenic stage 3 and could be detected in non-thermogenic stage 1. However, comparing to thermogenic receptacles, the AOX protein in non-thermogenic leaves had a much weaker band which was almost undetectable. In contrast, pUCP had a very low content in thermogenic receptacles and a high content in non-thermogenic leaves. Gray value analysis showed significant differences between the thermogenic stage and the non-thermogenic stage or tissue (Figure 5E). AOX was abundant in the thermogenic stage of receptacles while pUCP was abundant in non-thermogenic leaves.

### 2.6. Protein-Protein Interaction (PPI) Network and Weighted Gene Co-Expression Network Analysis (WGCNA)

To reveal protein regulatory networks, a PPI network analysis of the proteins in cluster 3 was performed using the STRING PPI database. Protein-protein relationships in the top 200 confidence intervals were used to construct the PPI network (Figure 6). Several highly connected proteins were identified. For example, CIT (citrate synthase, mitochondrial, NNU_11668-RA), a protein involved in the TCA cycle, had the most edges and represented the top hub protein (highly connected protein) in cluster 3. Moreover, succinate dehydrogenase (ubiquinone) iron-sulfur subunit 2, mitochondrial, NNU_03292-RA (SDH2-2), dihydrolipoyl dehydrogenase 1, mitochondrial, NNU_24547-RA (LPD1), and isocitrate dehydrogenase (NADP) isoform X1, NNU_21884-RA (IDH) were the other hub proteins with no less than 10 edges.

WGCNA [37,38] was performed to reveal protein modules related to the thermogenic phenotype and developmental stages. As a result, ten modules with different expression patterns were identified (Figure 7A,B, Appendix A, and Appendix A). Based on the criteria (Pearson r ≥ 0.8 or r ≤ −0.8, *p*-value ≤ 0.05), we found that the red module of 78 proteins was highly correlated with the thermogenic phenotype (Figure 7C). The co-expressed proteins network of the red module was visualized, of which several hub proteins were identified (Figure 7D). For example, the top hub protein was glucose-1-phosphate adenylyltransferase small subunit, chloroplastic/amyloplastic (APS1, NNU_20629-RA) involved in the process of starch biosynthesis. Then, KEGG function enrichment analysis showed that metabolic processes and fatty acid degradation were the main enriched pathways (Figure 7E, Appendix A). We also found that ubiquinone and other terpenoid-quinone biosynthesis were enriched, of which ubiquinone was an essential component in the mitochondrial electron transport chain.

## 3. Discussion

### 3.1. The Low-Correlation Omics Data Showed Similar Functions That Related to Thermogenesis

In the present study, we identified 640 DAPs in cluster 3 with high abundance and 429 DAPs in cluster 5 with low abundance during thermogenesis of *N. nucifera*. A previous study from our laboratory has identified 3054 thermogenesis-related genes in *N. nucifera* at the transcript level [32], of which 1795 differentially expressed genes were highly expressed and 1259 DEGs were low-expressed genes during thermogenesis. As far as these DEGs/DAPs related to thermogenesis were concerned, only 155 DEGs/DAPs with high abundance and 57 DEGs/DAPs with low abundance during thermogenesis were co-expressed at both transcription and protein levels (Appendix A). Therefore, the transcriptome data and proteome data were less correlated. However, the functional annotation analysis at the two omics levels suggested that the low-correlation omics data showed similar functions that related to thermogenesis. The DEGs/DAPs highly expressed during thermogenesis at the two omics levels were enriched in the KEGG pathways related to the tricarboxylic acid cycle, carbon metabolism, 2-carboxylic acid metabolism, glycolysis/gluconeogenesis, lysine degradation, and other metabolic processes. Moreover, in enriched GO terms of cellular component, mitochondrion was one of the most enriched terms. These results indicated that thermogenesis was related to respiratory metabolism and mitochondrial function, which was consistent with the study of gene expression profiles in *S. renifolius* [30]. Previously studies have suggested that carbohydrates, especially starch, were the respiratory substrates for floral thermogenesis in *N. nucifera* [20,25]. KOG annotation of cluster 3 showed that these proteins with high abundance during thermogenesis were concentrated in energy production and conversion as well as carbohydrate transport and metabolism, and starch and sucrose metabolism was one of the most significantly enriched KEGG pathways (Figure 3). Moreover, APS1, a protein involved in the synthesis of starch, was identified as a hub protein of the co-expressed proteins network (Figure 7D). These results indicated that the enhancement of cell respiration metabolism and the consumption of carbohydrate substrates help the flowers of *N. nucifera* to generate enough heat during anthesis. Similarly, in the present study, the DEGs/DAPs with low abundance during thermogenesis were significantly enriched in the photosynthetic metabolic pathway. The function of the receptacles during the thermogenic periods was mainly to generate heat. After thermogenesis was over, the color of the receptacles gradually changed from yellow to green with its function changing to photosynthesis [36].

### 3.2. The Protein Expression Patterns of AOX and pUCP between Thermogenic and non-Thermogenic Tissues were Completely Different

Although the COX flux did not change significantly in the receptacles of *N. nucifera* during the five stages, it implied that pUCP might not play a role in the thermogenesis of *N. nucifera* [20]. However, there was no direct evidence showing the abundance of pUCP in receptacles tissues. Grant et al. [20] tried to detect pUCP in the receptacles of *N. nucifera* by immunoblotting, but no band was detected. In the present study, the abundance of pUCP was determined by Western blot assays in the receptacles at the five stages (Figure 5). The abundance change of pUCP was similar to AOX, but there was almost no significant change in comparison with AOX (Figure 5C). The insignificant change in pUCP abundance was consistent with the insignificant change in COX flux. Then, in the comparison between non-thermogenic and thermogenic tissues, pUCP accumulated in non-thermogenic leaves, while AOX accumulated in thermogenic receptacles (Figure 5D,E). Therefore, the expression patterns of AOX and pUCP were opposite between non-thermogenic and thermogenic tissues. Coincidentally, the same situation was detected in the spadices of thermogenic *Symplocarpus renifolius* and non-thermogenic *Lysichiton camtschatcensis*, which are closely related to skunk cabbages in morphology and phylogeny [29]. AOX accumulated in thermogenic *S. renifolius* while pUCP accumulated in non-thermogenic *L. camtschatcensis*. These results suggested the important role of AOX, rather than pUCP, in the thermogenesis of these thermogenic tissues. Taken together, we have provided more direct evidence that pUCP is unlikely to play a role in thermogenic receptacles. In addition, AOX accumulated abundantly in the receptacles and could even be detected in non-thermogenic stage 1 (Figure 5D), which indicated that AOX protein was specifically abundant in thermogenic tissue.

### 3.3. TCA Cycle Metabolism, Fatty Acids Degradation, and Ubiquinone Biosynthesis Participate in Regulating Floral Thermogenesis of N. nucifera

TCA cycle intermediate metabolites can affect the synthesis of AOX protein [39]. Research has reported that an increase in the content of citrate (1–10 mM) caused a dramatic increase in *AOX1* mRNA, AOX protein, and AOX respiratory capacity in tobacco suspension cells [17]. Citrate could induce a stable increase in *AOX1* mRNA, even at a condition close to physiological concentration (0.1 mM) or at a high concentration of 20 mM that affected cell viability [40]. In *Arabidopsis*, perturbations in cellular concentrations of citrate also increased *AOX1a* transcript and had a major impact on nucleus-encoded transcript abundance [41]. In the present study, the TCA cycle was the most significantly enriched KEGG pathway, and several hub proteins identified by the PPI network were key enzymes of the TCA cycle (Figure 6). The top hub protein was citrate synthase, which catalyzed the irreversible condensation reaction of oxaloacetate and acetyl-CoA in the TCA cycle to form citrate [42]. The reaction catalyzed by citric synthase is the initial and key step of the TCA cycle, which can affect the metabolic rate of the entire cycle [43,44]. Our results suggested that the high abundance of citrate synthase during thermogenesis might promote the increase of citrate content. Increased citrate could be a physiological signal between mitochondrial metabolic state and nuclear gene expression, which then affects the expression of the *AOX* gene [45]. Although the mechanism that mediates this physiological signal remains to be discovered [46], it is clear that changes in the TCA cycle metabolism during thermogenesis affect the expression of the nuclear-encoding *AOX* gene, thereby contributing to thermogenesis.

Our WGCNA analysis obtained a protein module related to the thermogenic phenotype. KEGG enrichment analysis of the proteins in this module showed that fatty acid degradation was one of the most enriched pathways in this module (Figure 7E). Since the increase in free fatty acid concentration inhibits AOX activity [47], and *N. nucifera* depends on the AOX-mediated alternative pathway for thermogenesis [20,26], the metabolic activity of fatty acid degradation occurring in the receptacles may facilitate the maintenance of normal activity of AOX, thereby contributing to heat production.

The redox state of ubiquinone and the absolute amount of ubiquinone could affect AOX activity [48,49]. It has been suggested that a low ubiquinone level might limit AOX activity [49]. In the present study, our KEGG enrichment analysis of the proteins in the red module also showed that ubiquinone biosynthesis was significantly enriched (Figure 7E). Thus, the biosynthesis metabolism of ubiquinone can also affect AOX activity, thereby regulating floral thermogenesis.

### 3.4. Putative Model for the Regulatory Networks Involved in the Regulation Mechanism of Floral Thermogenesis in N. nucifera

To summarize, based on the above results and discussion, we proposed a model that demonstrated the regulatory networks involved in the regulation mechanism of floral thermogenesis in *N. nucifera* (Figure 8). During the thermogenic stages of development, mitochondria first respond to ambient temperature changes, and thus the functional status of mitochondria that includes TCA cycle metabolism is altered. Some TCA intermediate metabolites are considered as physiological signals to induce the expression of nuclear-encoded *AOX* genes [45,46]. The signals of mitochondria transmitted to the nucleus to regulate the expression of nuclear-encoded mitochondrial proteins are called mitochondrial retrograde regulation (MRR) [50]. The induction of the *AOX* gene increases the content of AOX protein, which is beneficial to thermogenesis. In addition, succinate as a TCA intermediate has been shown to stimulate the activity of AOX in *N. nucifera* [27]. At the same time, the enhancement of the metabolic activity of fatty acid degradation and ubiquinone synthesis also contributes to the increase of AOX activity, thereby facilitating thermogenesis. In fatty acid degradation, 3-hydroxyacyl-CoA dehydrogenase (HAD, NNU_21424-RA), which was identified as a hub protein (Figure 7D), participates in the β-oxidation of fatty acids that is the main pathway of fatty acid degradation. Then, the increase in the metabolic activity of starch and sucrose provides essential respiratory substrates for the maintenance of thermogenesis. In summary, the regulation mechanism of floral thermogenesis in *N. nucifera* involves complex regulatory networks. In addition to the above mechanisms, changes in ROS homeostasis may also act as retrograde signals to affect the expression of *AOX* genes in *N. nucifera*. This requires further research to explore more comprehensive regulatory networks.

## 4. Materials and Methods

### 4.1. Plant Materials

A cultivar ‘E Zilian 1′ of *N. nucifera*, which was provided by the Institute of Vegetables, Wuhan Academy of Agricultural Sciences, Wuhan, Hubei, China, was used for DIA-based quantitative proteomic analysis. The cultivar had a new plant variety right (CNA20130464.0) granted by the Ministry of Agriculture of the People’s Republic of China. The materials were preserved and cultivated in Wuhan National Germplasm Repository for Aquatic Vegetables (30°12′ N, 111°20′ E), with the preservation number V11A0692. As previously described, the development of *N. nucifera* flowers was divided into five sequential periods: firstly, the flowers were still small green buds at stage 1 and the flower buds became larger with pink tips and were close to blooming at stage 2; then, at stage 3, the full pink petals were half-open, showing mature pistils and immature stamens closely appressed to the receptacles; next, at stage 4, the petals were open horizontally and the stamens with mature pollen were spread out; finally, the petals and stamens withered at stage 5 (Appendix A) [20,32]. The temperature of each receptacle was measured with a needle thermocouple and a RDXL4SD digital thermometer (OMEGA, USA) (Appendix A). The receptacles of these five periods were collected, with three biological replicates in each period, immediately frozen in liquid nitrogen, and stored at −80 °C.

### 4.2. Protein Preparation and Digestion

Protein preparation was performed as follows: (1) Weigh approximately 0.4 g of each sample into a 2 mL centrifuge tube. Add 5% polyvinylpolypyrrolidone (PVPP) powder, a 5 mm magnetic bead, 1 mL lysis buffer (7 M Urea, 2 M Thiourea, 0.2% SDS, 20 mM Tris-Cl, pH 8.0), Phenylmethylsulfonyl fluoride (PMSF) with a final concentration of 1 mM and 2 mM Ethylenediaminetetraacetic acid (EDTA) to each tube, then vortex and incubate for 5 min, and add dithiothreitol (DTT) with a final concentration of 10 mM. (2) Shake the above mixture with a tissue grinder for 2 min (power = 50 Hz, time = 120 s) and centrifuge at 25,000 g for 20 min at 4 °C. (3) Transfer the supernatant into new centrifuge tubes and add DTT at a final concentration of 10 mM, followed by a water bath at 56 °C for 1 h. (4) After returning to room temperature, add iodoacetamide at a final concentration of 55 mM in the darkroom and incubate for 45 min. (5) Later, add four times the sample volume of cold acetone and keep at −20 °C for 2 h, followed by centrifugation at 25,000 g for 20 min at 4 °C. Repeat this step two to three times until the supernatant is colorless. (6) After the supernatant is carefully discarded, add a 5 mm magnetic bead and an appropriate amount of lysis buffer without SDS to the precipitate of each tube. (7) Shake the precipitate mixed with the lysis buffer using a tissue grinder (power = 50 Hz, time = 120 s) and centrifuge at 25,000 g for 20 min at 4 °C to obtain the supernatant, which is used for quantification, and then store at −80 °C in a refrigerator. Protein concentration was measured using the Bradford assay with bovine serum albumin as the standard [51]. SDS-PAGE and Coomassie brilliant blue R-250 staining assay were taken to evaluate the quality of the samples.

For protein digestion, the protein solution was diluted four-fold to reduce the concentration of urea before digestion. Each protein sample (100 μg) was digested for 4 h with trypsin at 37 °C (trypsin: protein = 1:40 (*v*/*v*), Hualishi Tech. Ltd., Beijing, China). Then, trypsin was added once more at the above ratio, and the hydrolysis was continued at 37 °C for 8 h. The enzymatically hydrolyzed peptides were desalted by Strata X column and dried under vacuum.

### 4.3. High pH Reverse Phase Fractionation

All samples were mixed equally (10 μg/sample) and diluted with 2 mL mobile phase A (5% acetonitrile (CAN) pH 9.8) and injected into Shimadzu LC-20AB liquid system (Shimadzu, Kyoto, Japan). A 5 um, 4.6 × 250 mm Gemini C18 column was used for liquid phase separation of the samples. The peptides were eluted at a 1 mL/min flow rate with a set of gradients: 5% buffer B (95% ACN, pH 9.8) for 10 min, 5% to 35% buffer B for 40 min, 35% to 95% buffer B for 1 min, buffer B maintained for 3 min and 5% buffer B equilibrated for 10 min. To collect the almost equivalent amounts of peptides from each pool, the elution peak was monitored at a wavelength of 214 nm and the components were collected every minute. Components were combined into a total of 10 fractions, which were then freeze-dried.

### 4.4. High-Performance Liquid Chromatography (HPLC)

The dried peptide samples were reconstituted with mobile phase A (2% ACN, 0.1% formic acid (FA)), centrifuged at 20,000 g for 10 min, and the supernatant was taken for injection. Separation was carried out with an UltiMate 3000 UHPLC system (Thermo Scientific, Waltham, MA, USA). The samples first entered the trap column to be enriched and desalted, and then were connected in series with a self-packing C18 column (150 μm inner diameter, 1.8 μm column material particle size, 25 cm column length), and were separated by the following effective gradient at a flow rate of 500 nL/min: 5% buffer B (98% ACN, 0.1% FA) for 0–5 min; 5% to 35% buffer B for 5–160 min; 35% to 80% buffer B for 160–170 min; 80% buffer B for 170–175 min; 5% buffer B for 176–180 min. The end of the nanoliter liquid phase separation was directly connected to the mass spectrometer.

### 4.5. Data-Dependent Acquisition Mass Spectrometry

The peptides separated by the liquid phase were ionized by the nanoESI source and then entered the tandem mass spectrometer Q-Exactive HF (Thermo Fisher Scientific, San Jose, CA, USA) for DDA mode detection. The main parameters were set as follows: Ion source voltage: 1.6 kV; MS scan range: 350–1500 *m*/*z*; MS resolution: 60,000; Maximal injection time (MIT): 100 ms; MS/MS collision type: Higher-energy collisional dissociation (HCD); Normalized collision energy (NCE): 28; MS/MS resolution: 30,000; MIT: 100 ms; Dynamic exclusion duration: 30 s. The start m/z for MS/MS was fixed to 100. Precursor for MS/MS scan satisfied: Charge range 2+ to 7+, top 20 precursors with intensity over 10,000. The automatic gain control (AGC) target was set to 3e6 for MS and 1e5 for MS/MS.

### 4.6. Data-Independent Acquisition Mass Spectrometry

DIA-MS was performed under the same LC system conditions as described above. The mass spectrometer was run under DIA mode and automatically switched between MS and MS/MS modes. The full scan was performed between 350–1500 *m*/*z* at 120,000 resolution, and the AGC target for the MS scan was set to 3e6 and the MIT was 50 ms. The full scan range (350–1500 *m*/*z*) was equally divided to 40 continuous windows for MS/MS scan with parameters set as follows: MS/MS collision type: HCD; MIT: Auto mode; MS/MS resolution: 30,000; Dynamic exclusion duration: 30 s; Stepped collision energy: 22.5, 25, 27.5; AGC target: 1e5.

### 4.7. Data Analysis

The raw data of DDA-MS were analyzed in the MaxQuant environment v.1.5.3.30 [52], employing the Andromeda search engine [53]. The MS/MS spectra were searched against the *N. nucifera* genome database LOTUS-DB (26641 entries, downloaded on 15 December 2018, http://lotus-db.wbgcas.cn) [54]. The search parameters were as follows: (1) Digestion enzyme: Trypsin with a maximum of two missed cleavages; (2) Minimal peptide length: 7; (3) Fixed modifications: Carbamidomethyl (C); and (4) Variable modifications: Oxidation (M) and Acetyl (Protein N-term). A false discovery rate (FDR) of 0.01 was set at the peptide and protein levels. The MaxQuant output file was used as a standard spectral library.

The raw data from DIA-MS were analyzed with the Spectronaut Pulsar 11.0 with default settings, which used the iRT peptides for retention time calibration [55]. Decoy generation was set to mutated, which is similar to scrambled but will only apply a random number of AA position swamps (min = 2, max = length/2). The false discovery rate (FDR) was estimated with the mProphet scoring algorithm [56] and set to 1% at the peptide and protein levels. Next, the R package MSstats [57] finished log2 transformation, normalization, and *q*-value (adjusted *p*-value) calculation. Differentially abundant proteins (DAPs) were also analyzed by MSstats, according to a fold change ≥ 2 and *q*-value < 0.05 (Student’s *t*-test) as the screening criteria.

### 4.8. Bioinformatic Analysis

We annotated the function of proteins based on the following databases: Nr (NCBI nonredundant protein sequences), Swiss-Prot (A manually annotated and reviewed protein sequence database), KOG/COG (Clusters of Orthologous Groups of proteins), and KEGG. GO annotation was performed against the Nr database using the Blast2GO program [58]. The proteins were mapped to different biological pathways according to functional categories in the KEGG pathway database (http://www.genome.ad.jp/kegg/pathway.html). GO terms and KEGG pathways enrichment analysis were performed based on the presence of two proteins and the results of a hypergeometric test (*p*-value ≤ 0.05 indicated statistical significance). PCA was completed by psych package in R. Time series analysis was performed using the Mfuzz R package [59]. The heat maps of gene expression and protein abundance were drawn by MeV software (version 4.9.0). PPI network was constructed using the STRING online tool (https://string-db.org/) and visualized with the Cytoscape software (version 3.6.1) [60]. WGCNA was performed with an online analysis platform (http://www.ehbio.com/Cloud_Platform/front/#/). The core parameters for constructing WGCNA network were as follows: Power = 12, minimal module size = 25, network type = signed, merge cut height = 0.2, deep split = 2.

### 4.9. Quantitative Real-Time PCR (qRT-PCR) Analysis

The total RNA was extracted as described by Zou et al. [32]. Next, 1 µg RNA sample of each stage was used for reverse transcription with the RevertAid First Strand cDNA Synthesis Kit (Thermo Scientific, Waltham, MA, USA), according to the manufacturer’s instructions. Then, qRT-PCR was performed with a StepOne Plus real-time PCR system (Applied Biosystems, Waltham, MA, USA) using a FastStart Universal SYBR Green Master Mix (Rox) (Roche, Germany). The reaction procedure was as follows: An initial denaturation at 95 °C for 10 min, followed by 40 cycles of 95 °C for 15 s, 60 °C for 1 min. The melting curve was determined for each primer pair to ensure specific amplification during the reaction process. Three replicates were made for each sample. The primers used in these experiments are shown in Appendix A. *NnEF1a* gene (GenBank accession number AB491177.1) was used as the internal control, according to Zhu et al. [61], and the relative expression levels were normalized and calculated using the 2^−ΔΔCt^ method [62].

### 4.10. Western Blot Analysis

Equal protein samples were separated on SDS-PAGE and transferred to the nitrocellulose membrane. The membrane was blocked for 1 h at room temperature in TBST buffer (25 mM Tris, 137 mM NaCl, 2.7 mM KCl, 0.05% Tween, pH 7.4) containing 5% non-fat milk, and was then washed three times with TBST buffer. Primary antibodies were diluted in TBST buffer containing 3% non-fat milk (AOX at 1:1000; pUCP at 1:2000; CAT at 1:1000) and incubated with the membrane for 2 h at room temperature with agitation or overnight at 4 °C. After being rinsed three times with TBST buffer, the membrane was incubated with the secondary antibody (diluted at 1: 6000) for 1 h at room temperature, and then washed in TBST buffer three times. Finally, the prepared membranes were incubated with WesternBright^TM^ ECL-HRP Substrate (Advansta, USA) for 1 to 3 min and put in a ChemiDoc^TM^ Imaging System (Bio-Rad, Hercules, CA, USA) to visualize signal intensities. Western blot assays were repeated at least three times. Grayscale value analysis was performed using ImageJ software (National Institutes of Health, Bethesda, MD, USA).

The antibodies used for Western blot assays in the present study were as follows: (1) primary antibody: Anti-AOX1/2 (Agrisera, Vännäs, Sweden), anti-UCP (Agrisera, Vännäs, Sweden), anti-CAT (Agrisera, Vännäs, Sweden), and plant actin monoclonal antibody (ImmunoWay Biotechnology, Plano, TX, USA); (2) Secondary antibody: HRP-conjugated goat anti-rabbit/mouse IgG secondary antibodies (Proteintech Group Inc., Rosemont, IL, USA).

### 4.11. Statistical Analysis

Statistical significance of the data was determined using the Student’s *t*-test to compare the two groups, and the one-way ANOVA test to compare multiple groups (*p* < 0.05).

## 5. Conclusions

In the present study, we reported the large-scale protein profiles related to floral thermogenesis through the next generation of label-free quantitative proteomics identification technology. A total of 6913 proteins were identified from receptacles at five different developmental stages, of which ten comparison groups with 3513 proteins were DAPs. Subsequently, we have identified 640 DAPs that were highly abundant during thermogenesis. The functional annotation of these proteins was mainly involved in pathways related to cellular respiration metabolism, such as the TCA cycle. Western blot analysis of AOX and pUCP further confirmed that pUCP was unlikely to participate in the floral thermogenesis of *N. nucifera*. Then, a protein module highly related to the thermogenic phenotype was identified by WGCNA analysis, which was mainly involved in metabolic processes, fatty acid degradation, and ubiquinone synthesis. In short, complex regulatory networks including TCA cycle metabolism, starch and sucrose metabolism, fatty acid degradation, and ubiquinone synthesis may participate in the regulation of floral thermogenesis in *N. nucifera*. The acquisition of the *N. nucifera* receptacle proteome will further enrich the research in the field of lotus proteomics and provide candidate proteins for future studies.

## Figures and Tables

**Figure 1 ijms-22-08251-f001:**
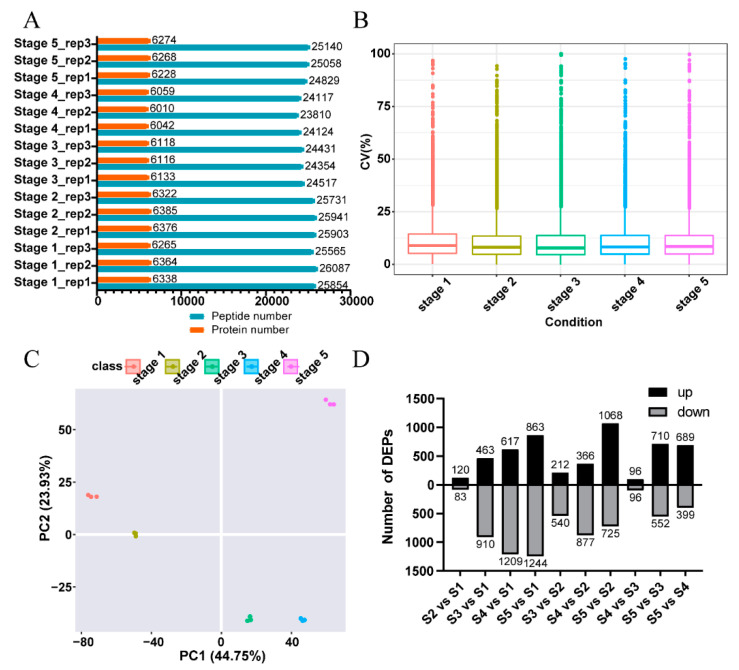
An overview of the quantitative proteomics analysis: (**A**) Statistics of the proteins and peptides identified in each sample. Each stage includes three replications. The ‘rep’ represents ‘replication’ on the y-axis. (**B**) Distribution of intra-group coefficient of variation (CV) for each sample. (**C**) Principal component analysis (PCA) of the quantified proteins for each sample. (**D**) Statistics of differentially abundant proteins (DAPs) in each comparison group. S1, S2, S3, S4, and S5 represent stage 1, stage 2, stage 3, stage 4, and stage 5, respectively.

**Figure 2 ijms-22-08251-f002:**
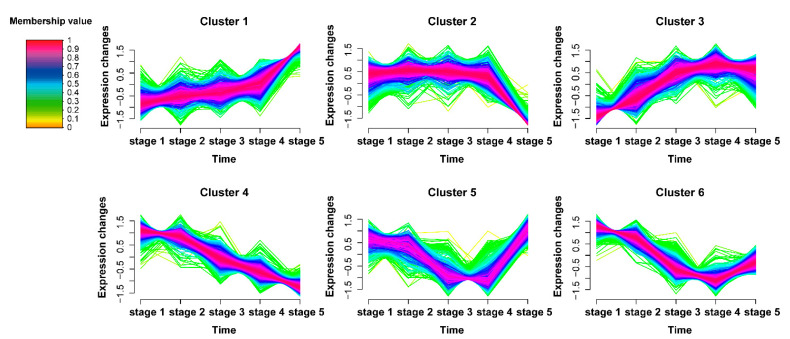
Time series analysis of DAPs: Time series analysis identified six distinct temporal patterns of proteins abundance using fuzzy c-means algorithm. The x-axis represents five developmental stages, while the y-axis represents normalized abundance changes of DAPs. The membership value represents the strength of the relationship between a protein and a corresponding cluster.

**Figure 3 ijms-22-08251-f003:**
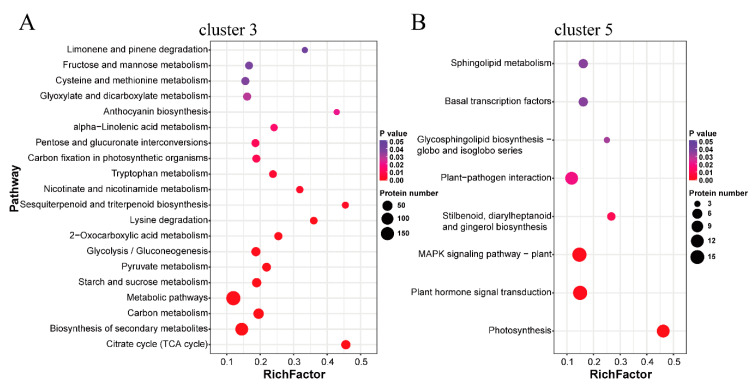
Kyoto encyclopedia of genes and genomes (KEGG) pathway enrichment analysis and Eukaryotic orthologous groups (KOG) annotation: (**A**,**B**) The main enriched KEGG pathways of cluster 3 and 5. The x-axis indicates the enrichment factor (RichFactor) which represents the number of DAPs annotated to each pathway divided by the number of all identified proteins annotated to the same pathway. A greater RichFactor indicates greater enrichment. The y-axis represents significantly enriched pathways with increasing significance from top to bottom.

**Figure 4 ijms-22-08251-f004:**
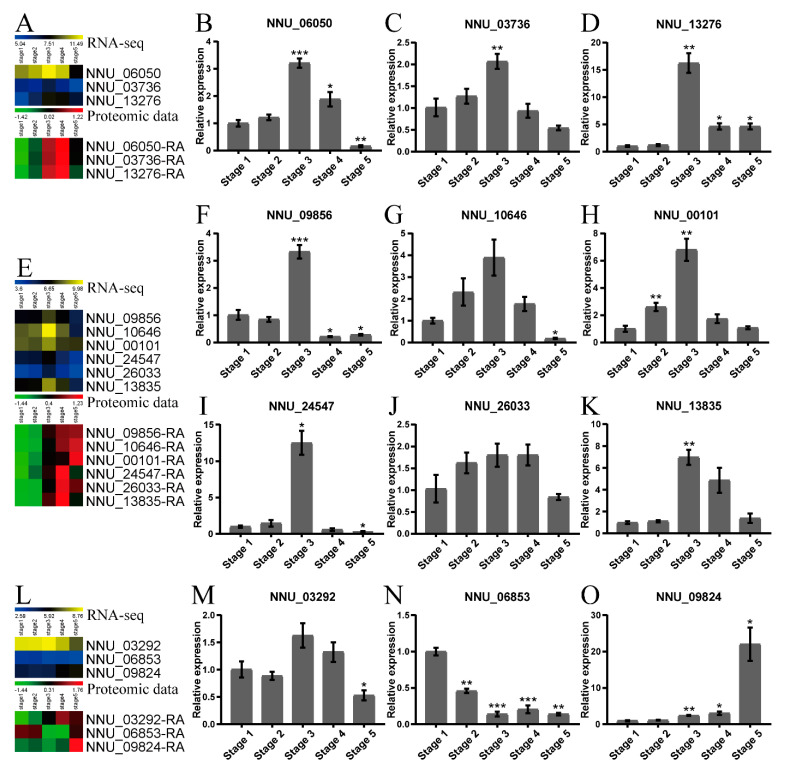
Validation of 12 DAPs by qRT-PCR: (**A**–**D**) Heatmap and relative transcript expression of AOX1a (alternative oxidase 1a, NNU_06050-RA) and pUCPs (plant uncoupling proteins, NNU_03736-RA and NNU_13276-RA). (**E**–**K**) Heatmap and relative transcript expression of NAD-ME1 (NAD-dependent malic enzyme 1, mitochondrial, NNU_09856-RA), IDH3 (Isocitrate dehydrogenase (NAD) regulatory subunit 3, mitochondrial, NNU_10646-RA), CYTC (Cytochrome c, NNU_00101-RA), LPD1 (Dihydrolipoyl dehydrogenase 1, mitochondrial, NNU_24547-RA), HXK1 (Hexokinase-1, NNU_26033-RA) and GAL1 (Galactokinase, NNU_13835-RA). (**L**–**O**) Heatmap and relative transcript expression of SDH2-2 (Succinate dehydrogenase (ubiquinone) iron-sulfur subunit 2, mitochondrial, NNU_03292-RA), probably inactive leucine-rich repeat receptor-like protein kinase IMK2 (NNU_06853-RA) and ATPD (ATP synthase subunit delta, chloroplastic, NNU_09824-RA, NNU_09824-RA). The heatmaps were used to visualize the transcript levels in the RNA-seq data and the protein levels in the proteomic data. The error bars indicate the standard deviation of three replicates. Asterisks indicate a significant difference as determined by one-way ANOVA using stage 1 as a control (* *p* < 0.05, ** *p* < 0.01, *** *p* < 0.001).

**Figure 5 ijms-22-08251-f005:**
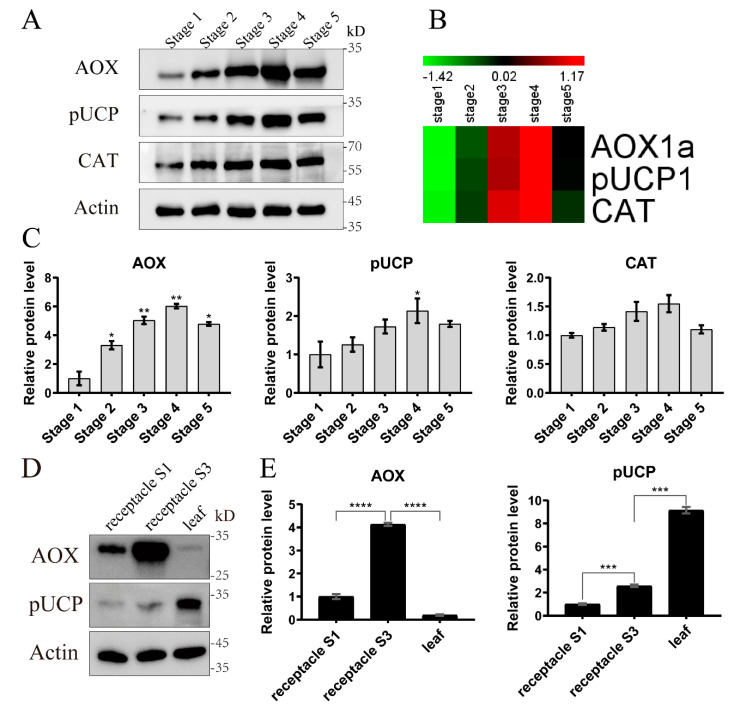
Western blot assays: (**A**) The expression patterns of AOX, pUCP, and catalase (CAT) validated by Western blot. (**B**) Quantitative proteomic data of AOX, pUCP, and CAT visualized by heatmap. (**C**) Grayscale value analysis of the results in Figure 5A using Image J. (**D**) The abundance of AOX and pUCP in different tissues of *N. nucifera*. (**E**) Grayscale value analysis of the results in Figure 5D. S1 and S3 represent stage 1 and stage 3, respectively. The error bars indicate the standard deviation of three replicates. Asterisks indicate a significant difference as determined by one-way ANOVA using stage 1 as a control in Figure 5C and ‘receptacle S3′ as a control in Figure 5E (* *p* < 0.05, ** *p* < 0.01, *** *p* < 0.001, **** *p* < 0.0001).

**Figure 6 ijms-22-08251-f006:**
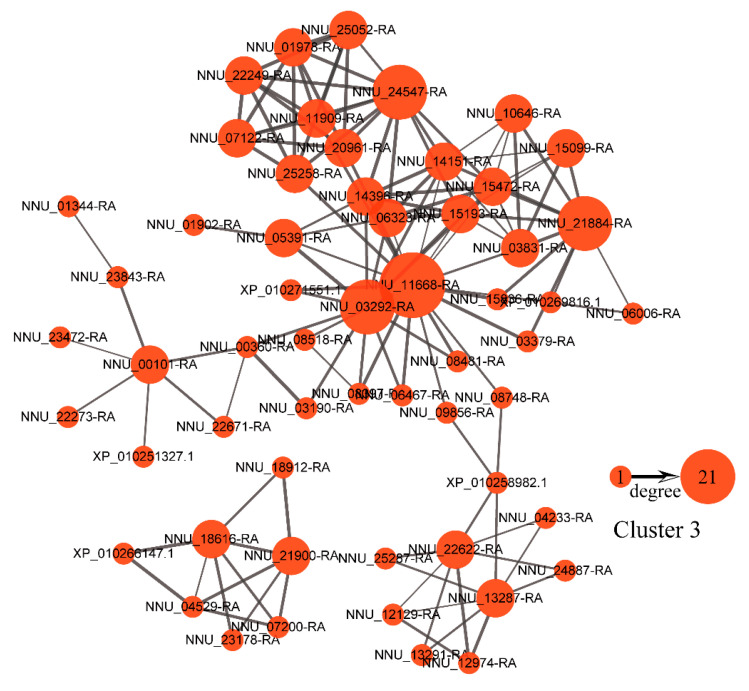
Protein-protein interaction (PPI) network of the proteins in cluster 3: The edge numbers of the proteins range from 1 to 21.

**Figure 7 ijms-22-08251-f007:**
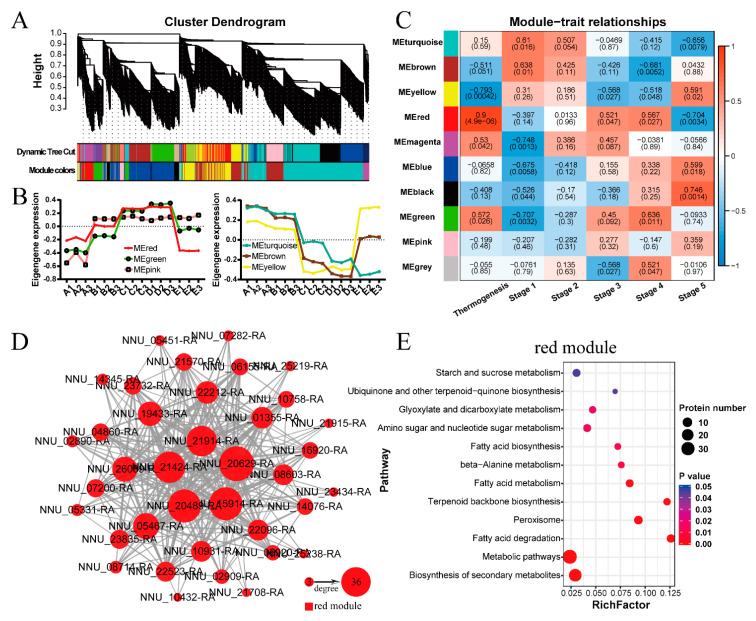
Weighted gene co-expression network analysis: (**A**) Hierarchical clustering dendrogram shows co-expression modules that are color-coded. ‘Dynamic Tree Cut’ represents initial modules. Module colors represent the final modules. Each branch in the hierarchical tree or each vertical line in color bars represents one protein. (**B**) Temporal expression patterns of different modules. Eigengene is the first principal component of a given module, which can be considered a representative of the protein profiles in a module. A, B, C, D, and E represent stage 1, stage 2, stage 3, stage 4, and stage 5, respectively. Each stage includes three replications. (**C**) Module trait correlation plot. Each row represents one module. Each column represents one trait attribute. The blue color represents negative correlation and the red color represents positive correlation. (**D**) The co-expressed proteins network of the red module with edge weight ≥ 0.52. The edge numbers of the proteins range from 3 to 36. (**E**) KEGG enrichment analysis of the proteins in the red module.

**Figure 8 ijms-22-08251-f008:**
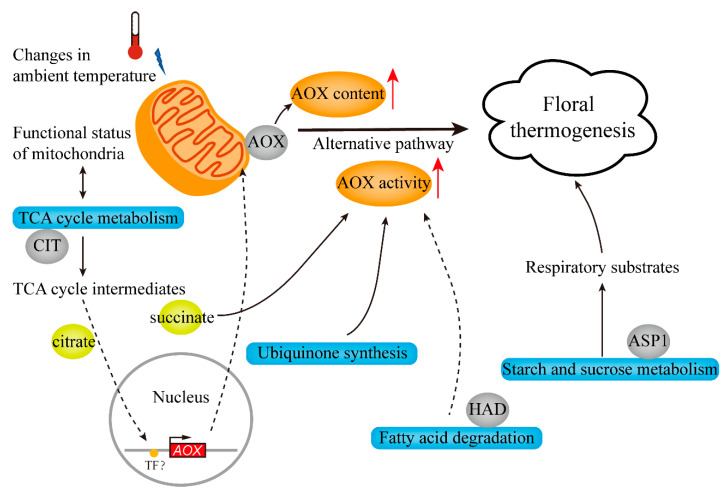
Putative model for the regulatory networks involved in the regulation mechanism of floral thermogenesis in *N. nucifera*. TCA, tricarboxylic acid; CIT, citrate synthase; HAD, 3-hydroxyacyl-CoA dehydrogenase; ASP1, glucose-1-phosphate adenylyltransferase small subunit.

## Data Availability

The mass spectrometry proteomics data have been deposited to the ProteomeXchange Consortium (http://proteomecentral.proteomexchange.org, 31 July 2021) via the iProX partner repository [63] with the dataset identifier PXD024269.

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
