# Peer review of "DIA-Based Quantitative Proteomics Reveals the Protein Regulatory Networks of Floral Thermogenesis in Nelumbo nucifera"

_ijms, 2021, doi:10.3390/ijms22158251_

Round 1

Reviewer 1 Report

Comments and suggestions for authors are available in the attached pdf file. 

Author Response

Question 1: (line 35) The developmental stages of flowers have not been explained in this manuscript. Although the stages have been introduced in the previous publication (Zou et al., 2020), they have to be mentioned somewhere in the introduction or materials and methods.

Answer 1: Thanks for your suggestion. We have added the description for developmental stages of flowers in the materials and methods.

Q2: (line 125) Fig.1: The labels on the "x" axis in the panel (D) in figure 1 ("S" in panel "D" versus "stage" in panels A-C) is not consistent with the rest of the labels.

Figure 1- panel A: The labels on Y axis needs to be re-written in a proper format.

A2: Thank you for pointing out the question. Due to the limited space of x-axis in Figure 1D, we use “S” to represent “Stage” for simplicity. We have explained this figure legend in the revised manuscript. As for the Y-axis labels in Figure 1A, the “rep” was used to represent “replication” for simplicity, we are sorry for not explaining it in the legend, we have also added it in the revised manuscript.

Q3: (line 141) Figure S4A is exactly similar to the one presented as Figure1D (The first 4 bars for up and down regulated proteins). It is confusing to present 2 same figures one as figure 1 and one as a supplementary figure.

A3: Thank you for your suggestion, we have deleted the corresponding supplementary figure in the revised manuscript.

Q4: (line143) Authors can highlight the 4 labels on X axis- figure 1, panel D, that represent the adjacent stages rather than using the supplementary figure.

A4: Thank you for your suggestion, we have deleted the corresponding supplementary figure and explained the labels representing the adjacent stages in the revised manuscript.

Q5: (line195) It seems the authors need to increase the X axis value for figure 3-panel B as the enrichment factor for photosynthesis is higher than 0.5.

A5: Thank you for pointing out the problem, we have increased the X axis value for figure 3-panel B in the revised manuscript.

Q6: (line203) Authors need to replace "increase in abundance of proteins" instead of "high expression". 

Authors rather use "protein accumulation" or "abundance" instead of "expression".

A6: Yes, according to your suggestion, we have used "protein accumulation" or "abundance" instead of "protein expression", and "high abundance" instead of "high expression", and changed “differentially expressed proteins (DEPs)” to “differentially abundant proteins (DAPs)” in the revised manuscript.

Q7: (line210) Authors have mentioned that they randomly selected 12 DEPs. However, the most interesting proteins that authors have focused in this study (AOX and pUCP) are on their list! Authors please explain this.

A7: Thank you for pointing out the problem. This was our presentation problem, which should actually be nine randomly selected DEPs and three proteins of interest. We have revised our presentation in the revised manuscript.

Q8: (245line) Why authors did not consider CAT for qPCR analysis?

A8: We found that CAT did not change significantly during the five developmental stages of N. nucifera flowers and was not a differentially expressed protein (DEPs). Therefore, we did not choose to perform QPCR analysis on it, but it could be used to validate the quantitative proteomic data in Western blot experiments.

Q9: (492line) The date of searching against the database?

A9: Thank you for pointing out the missing information. The search time is December 15, 2018. We have added the information in the revised manuscript.

Q10: (494line) Number of trypsin missed cleavage(s)? One or two?

A10: The maximum missed cleavages were set to two. We have added the information in the revised manuscript.

Reviewer 2 Report

The manuscript “DIA-based Quantitative Proteomics Reveals the Protein Regulatory Networks of Floral Thermogenesis in Nelumbo nucifera” submitted by Yueyang Sun and coworkers describes a quantitative proteomics study of floral thermogenesis of N. nucifera. In this work the authors applied data-independent acquisition (DIA)-based quantitative proteomics technology to evaluate five different stages of development of the floral tissue in order to understand the molecular details of thermogenesis in this plant. According to their findings, the authors claim that 640 proteins are highly expressed during floral thermogenesis, with a particular representation of proteins from the TCA cycle. In addition, the authors also performed qRT-PCR and western-blot in order to validate the results derived from proteomic experiments. Using these approaches, they verified that AOX specifically expressed in the receptacles during thermogenesis. Overall, the is an interesting manuscript, which is well presented and organized. In my opinion, the manuscript is suitable for publication after a minor revision.

Minor comment:

Lines 107-108: The authors claim that this proteomic study will reveal the “molecular basis of floral thermogenesis in N. nucifera”. Overall, the authors mention that AOX participate in the mechanism of thermogenesis. However, I would suggest the authors to include an additional Figure or scheme in which they present a complete overview of all the molecules and mechanisms involved in the mechanisms of thermogenesis that takes place in this plant.

Author Response

Q1: Lines 107-108: The authors claim that this proteomic study will reveal the “molecular basis of floral thermogenesis in N. nucifera”. Overall, the authors mention that AOX participate in the mechanism of thermogenesis. However, I would suggest the authors to include an additional Figure or scheme in which they present a complete overview of all the molecules and mechanisms involved in the mechanisms of thermogenesis that takes place in this plant.

A1: Thank you for your suggestion. We have added an additional Figure 8 in the discussion section of the revised manuscript. This figure is a putative model presenting the regulatory networks involved in the regulation mechanism of floral thermogenesis.